# How faculty define quality, prestige, and impact of academic journals

Esteban Morales[1]*, Erin C. McKiernan[2], Meredith T. Niles[3], Lesley Schimanski[4], Juan Pablo Alperin[4]*

1 University of British Columbia, Vancouver, Canada, 2 Universidad Nacional Autónoma de México, Mexico City, Mexico, 3 University of Vermont, Burlington, Vermont, 4 Simon Fraser University, Burnaby, Canada

* esteban.morales@ubc.ca (EM); juan@alperin.ca (JPA)

## Abstract

Despite the calls for change, there is significant consensus that when it comes to evaluating publications, review, promotion, and tenure processes should aim to reward research that is of high "quality," is published in "prestigious" journals, and has an "impact." Nevertheless, such terms are highly subjective and present challenges to ascertain precisely what such research looks like. Accordingly, this article responds to the question: how do faculty from universities in the United States and Canada define the terms quality, prestige, and impact of academic journals? We address this question by surveying 338 faculty members from 55 different institutions in the U.S. and Canada. While relying on self-reported definitions that are not linked to their behavior, this study's findings highlight that faculty often describe these distinct terms in overlapping ways. Additionally, results show that marked variance in definitions across faculty does not correspond to demographic characteristics. This study's results highlight the subjectivity of common research terms and the importance of implementing evaluation regimes that do not rely on ill-defined concepts and may be context specific.

## Introduction

Although faculty work involves a wide range of activities and priorities [1–3], evidence confirms that faculty, even those at teaching and public institutions, believe that research is the most highly valued aspect of their work [4–6]. It is therefore unsurprising that despite significant efforts to broaden how university faculty are assessed, research continues to be treated as the main component of a professor's job [6–8] and, accordingly, research assessment features prominently in how faculty are evaluated for career advancement. Yet, despite its perceived importance and critical role, there is still debate about how to evaluate research outputs.

Research assessment, which is codified for faculty career progression in review, promotion, and tenure (RPT) processes, is itself a controversial topic that has been subject to much debate over the years [c.f. 9]. Critics have argued that current assessment practices overemphasize the use of metrics—such as the Journal Impact Factor (JIF) [10]—and fail to recognize new and evolving forms of scholarship, such as datasets and research software [11–14], fail to encourage

**Data Availability Statement:** Survey responses can be found at the following publication: Niles, Meredith T.; Schimanski, Lesley A.; McKiernan, Erin C.; Alperin, Juan Pablo, 2020, "Data for: Why we publish where we do", https://doi.org/10.7910/

DVN/MRLHNO, Harvard Dataverse, V1 Data regarding RPT documents can be found at the following data publication: Alperin, Juan Pablo; Muñoz Nieves, Carol; Schimanski, Lesley; McKiernan, Erin C.; Niles, Meredith T., 2018, "Terms and Concepts found in Tenure and Promotion Guidelines from the US and Canada", https://doi.org/10.7910/DVN/VY4TJE, Harvard Dataverse, V3, UNF:6:PQC7QoilolhDrokzDPxxyQ== [fileUNF].

**Funding:** Funding for this project was provided to JPA, MTN, ECM, and LAS from the Open Society Foundations (OR2017-39637). The funders had no role in study design, data collection and analysis, decision to publish, or preparation of the manuscript.

**Competing interests:** MTN is a member of the board of directors of The Public Library of Science (PLOS). This role has in no way influenced the outcome or development of this work or the peer-review process, nor does it alter our adherence to PLOS ONE policies on sharing data and materials.

reproducible science [15], or worse, that current approaches encourage unethical practices, such as gift authorship, p-value hacking, or result manipulation [16, 17]. These discussions point to different perspectives regarding what constitutes research that is worthy of being celebrated and rewarded.

Despite the calls for change and the differences of opinions, there is significant consensus that when it comes to evaluating publications, the RPT process should aim to reward research that is of high "quality" and has "impact" [9]. However, such terms are highly subjective, and present challenges to ascertain precisely what such research looks like and where it ought to be published. Furthermore, their subjectivity presents additional challenges for comparing research and individuals, as is regularly done during the RPT process. The use of these subjective concepts, and others like them, may serve primarily as "rhetorical signalling device[s] used to claim value across heterogeneous institutions, researchers, disciplines, and projects rather than a measure of intrinsic and objective worth" [18].

Others have previously noted the lack of clear definitions surrounding many of the terms and concepts used in research assessment [18–21]. Without definitions, individuals and committees are bound to apply different standards which inevitably leads to inequities in how faculty and research are evaluated. If, as Hatch [20] suggests, "most assessment guidelines permit sliding standards," then the definition used in any given situation can easily shift depending on whose work is being assessed in ways that allow (or even encourage) biases to creep into the evaluation process (be they conscious or unconscious). Even if individuals are consistent and unbiased in how they apply their own definitions, there is rarely agreement in assessments between academics, even from those of the same discipline. Moore and colleagues [18] point to such conflicting assessments as the clearest example of a lack of any agreed upon definition of 'excellence'—an argument that can easily be extended to other terms and concepts commonly used in research assessment.

The known pernicious effects of using ill-defined criteria has resulted in calls to "swap slogans for definitions" [20] and for greater "conceptual clarity" [22] in research assessment. To aid in this effort, this study focuses on faculty's perception of academic journals as they carry significant weight in current RPT processes, especially in universities in the United States and Canada (Niles et al., 2020). Accordingly, this study addresses the question: how do faculty understand the concepts of 'quality', 'prestige', and 'impact' as they pertain to academic journals?

## Previous research

The current study builds on our previous work to study the documents related to RPT processes at institutions in the United States and Canada [10, 23, 24], as well as similar work by others in this area [25–27]. In one of our previous studies [23], we reported that nearly 60% of institutions overall and nearly 80% of research-intensive universities mentioned 'impact' in their RPT documents. Using the same data and analysis, but previously unreported, we also found that 73% of institutions in our sample and 89% of research-intensive universities mentioned 'quality' in their RPT documents, and that those percentages were 29% and 47% for those that mentioned 'prestige.' That is to say, there was a high prevalence of such concepts in academic evaluations, but a closer reading of these instances shows that few of these documents gave a clear definition of what these terms meant or how they were to be measured.

Despite the frequent use of these terms in relation to research assessment, it is difficult to know how faculty understand them, especially in relation to how they will assess others, and in how they understand that they will be assessed. A survey by DeSanto and Nichols [28] found that ". . .a significant number of faculty [are] unsure of their department's RPT expectations

for demonstrating scholarly impact" (pg. 156). Results from that same study show there is substantial disagreement among faculty as to how impact should be measured and evaluated, with many pushing for traditional journal-level metrics like the Journal Impact Factor (JIF) and a small percentage favoring new article-level metrics and altmetrics. Similarly, there is disagreement as to how research quality should be measured, with little evidence suggesting that it can be assessed through citation-based metrics like the JIF [29]. In lieu of an objective measure of quality, it has become common to use the perceived prestige of the publication venue (i.e., the journal where an article is published) as a proxy. To confound things further, prestige is itself sometimes associated with the JIF [e.g., 30], even while the association of the JIF with both quality and prestige has been heavily criticized, most notably in The San Francisco Declaration on Research Assessment [31, 32], the HuMetricsHSS Initiative [33] and in the Leiden Manifesto [34].

This interplay between the JIF, quality, prestige, and impact and how it features in research assessment is also evident when faculty make decisions about where to publish their research. A recent faculty survey [35] shows that nearly 80% of faculty report the JIF as one of the most important factors influencing their decisions on where to publish, something echoed in our own survey findings [24]. In some instances, faculty are guided by librarians to use the JIF to determine the prestige of a given journal [30] but, according to the same Ithaka survey, the majority of faculty make such decisions based on their own perceptions of the quality, prestige, and potential impact of a given journal. As the report states: "less than 20% of respondents reported they receive help determining where to publish to maximize impact, and assessing impact following publication" [35]. Further complicating our understanding of faculty's understanding and use of these concepts is the role that demographic characteristics—such as gender and career stage—significantly impact how scholarship is perceived and practiced [36, 37]. In our own previous work [24], for example, we found that women are more likely to publish fewer articles than men but often assign more importance to the number of publications than their male peers.

These surveys show that, when it comes to making decisions about where to publish, faculty see an interplay between the notions of quality, prestige, and impact and have themselves linked these to metrics like the JIF, although their precise understanding of these terms remains unclear. To some extent, these interconnected concepts have been codified in the RPT guidelines and documents that govern academic careers. As noted above, in previous work we uncovered the high incidence of the terms *quality* and *impact* in these documents [23] and, in another study, we uncovered that the JIF and related terms are found in the RPT documents of 40% of R-type institutions, and the overwhelming majority of those mentions support their use [10]. Similarly, Rice et al. [26] found mentions of the JIF in nearly 30% of the RPT guidelines from several countries, and also found support for the measure's use. Moreover, we found that although it is not always stated what the JIF is intended to measure, 63% of institutions that mentioned the JIF in their documents had at least one instance of associating the metric with quality, 40% had at least one mention associating it with impact, and 20% with prestige [10]. These results are in stark contrast to a number of studies showing that JIF has little or nothing to do with research quality [38–41].

The complexity of these terms, intertwined with their application in faculty behavior and promotion decisions, demonstrate a need to further understand how faculty themselves perceive these terms. Their persistent use in both publication decisions and in research assessment indicates their importance and the ambiguities, regardless of their reason for being, suggest that further study is needed. As such, we sought to answer the question: how do faculty from universities in the United States and Canada define the terms quality, prestige, and impact?

## Methods

To conduct this study, we sent an online survey using SurveyMonkey to 1,644 faculty members from 334 academic units from 60 universities from Canada and the United States. As described in greater detail in Niles et al. [24], we created this contact list based on a random sample of universities from which we have previously collected RPT guidelines. Faculty were invited to participate in a survey between September and October 2018. Ethics approval was provided through Simon Fraser University under application number 2018s0264. The study was not pre-registered. Written consent was obtained prior to data collection. We received responses from 338 faculty (21%) from 55 different institutions. Of these, 84 (25%) were faculty at Canadian institutions and the remaining 254 (75%) were from the United States; 223 (66%) were from R-Type institutions, 111 (33%) from M-Type institutions, and 4 (1%) from B-Type institutions. Full methodological details and demographic reporting of respondents can be found in Niles et al. [24].

In this paper, we present a detailed analysis of the responses to the question "*In your own words, how would you define the following terms, sometimes used to describe academic journals?*" The terms included in this question were *high quality*, *prestigious* and *high impact*. Of the 338 responses, 249 (74%) responded to this set of open-ended questions, for an effective response rate of 15% of the 1,644 invitations that were sent. We analyzed the responses using open-coding and constant comparison [42]. To achieve this, we first organized all the responses into segments—sentences, or parts of sentences, that convey a single idea. We then assigned codes to these segments, grouping those that represent the same idea or creating new codes when a new idea appeared. Each response could contain one or multiple segments, each of which could be coded differently, allowing for a single response to have multiple codes. This process continued until we developed a codebook that included the name of the codes, descriptions for each and examples, as shown in Tables 1–3 (below, in the results section).

In order to determine the inter-rater reliability of the codebook, two researchers independently coded the same randomly chosen set of 20 responses for each of the three terms, and compared the results using NVivo 12, which resulted in adjustments to the codebook and finally an overall Kappa value of 0.87 [43]. After having a good result in the inter-rater reliability test, all the responses were coded by one of the authors of this study (EM). Finally, the results of the open-coding process were analyzed by running Chi-Square tests upon different variables in Excel, in order to see the variation of the definitions in the light of different categories.

## Results

We present the results in three parts: first, we describe the results of the open-coding process for the three terms, as the codes themselves capture the definitions used by respondents; second, we analyze the frequencies of each code in relation to respondent's demographic information; and, finally, given the high incidence of the JIF in faculty definitions of the three terms, we explore the relationship between faculty definitions and the presence of the JIF in the RPT guidelines at the faculty member's institution and academic unit.

### Defining quality, prestige and impact

The analysis of the definitions provided for *high quality* resulted in 295 segments. These segments were categorized into five groups: *Impact factor and Metrics*, *Value*, *Readership*, *Reputation*, and *Review Process*. Table 1 provides the description of each group, as well as some examples for each of them.

**Table 1. Categories identified in the definitions of high quality.**

| Category | Description of the category | % of responses | Examples |
|---|---|---|---|
| Impact factor and metrics | Established measurement for the journals by the articles that are published, and the citation generated | 11.5% | • High journal ranking [373] |
| | | | • Journal impact factor [500] |
| Value | Quality and applicability of the articles published in the journal, including its scientific rigor and contribution to the field | 35.3% | • Quality of the research being presented [582] |
| | | | • Publishes well-researched, innovative articles [703] |
| Readership | Focuses on how much the published work is read by academic and non-academic people | 2.4% | • Large readership [62] |
| | | | • Is widely read [621] |
| Reputation | Influence and recognition of all the elements related to the journal, such as the editorial board, the scholars who publish or the journal itself | 8.5% | • Well-established, well-known editorial board [568] |
| | | | • name recognition of journal [162] |
| Review process | Elements related to the process of reviewing the articles that are published, such as editors, feedback and rejection rate. | 42.2% | • Peer-reviewed publication [1610] |
| | | | • Rigorous, reviewed by top reviewers in the field [600] |

Name, description, percentage of responses, and examples of the categories found in participants' definitions of High Quality. Numbers in square brackets represent anonymized identification of participants.

The result of this coding process shows that faculty most commonly define high quality academic journals based on the review process, referring to the perceived rigor of the process of evaluating, gatekeeping and editing academic articles for the journal (e.g., "rigorous review and selection of articles to publish" [848]). Another common view on what determines the quality of an academic journal is related to the perceived value of the articles published within it, including how consequential they are for the field, the quality of the writing, and the methodological standards of the research (e.g., "Good methodology, quality writing" [1624]).

**Table 2. Categories identified in the definitions of prestigious.**

| Category | Description of the category | % of responses | Examples |
|---|---|---|---|
| Impact factor and metrics | Established measurement for the journals by the articles that are published, and the citation generated | 18.3% | • Highly ranked [561] |
| | | | • Super impact factor and circulation [761] |
| Value | Scientific rigor and applicability of the articles published in the journal. | 12.2% | • Field changing, important, correct [488] |
| | | | • Usually very good quality [1351] |
| Readership | Focuses on how much the published work is read by academic and non-academic people | 4.6% | • Widely read [1171] |
| | | | • Large reading audience [86] |
| Relation to associations | Relation of the journal to organizations that support its operation. | 4.2% | • Association sponsored [812] |
| | | | • Affiliated with a widely recognized organization [577] |
| Reputation | Recognition of the journal itself, the authors of the articles published in the journal or the authors citing the journal. | 42.7% | • High name recognition [581] |
| | | | Held in high regard by researchers [214] |
| Review process | Elements related to the process of reviewing the articles that are published, such as editors, feedback, and rejection rate | 17.9% | • Expert peer-review [388] |
| | | | • Hard to get accepted for publication [365] |

Name, description, percentage of responses, and examples of the categories found in participants' definitions of Prestige. Numbers in square brackets represent anonymized identification of participants.

**Table 3. Categories identified in the definitions of high impact.**

| Category | Description of the category | % of responses | Examples |
|---|---|---|---|
| Impact factor and metrics | Established measurement for the journals by the articles that are published, and the citation generated | 49.2% | • Interplay between Impact Factor and number of cites per year [500]<br>• Use of impact factor to identify a journal's worthiness [862] |
| Impact on academia | Relevance and influence of the articles on future research | 16.0% | • Immediately impacting the next work to be published [583]<br>• Influences a lot of other researchers [391] |
| Impact outside academia | Impact on practices and public policies by the articles, as well as replicability on media outlets | 10.7% | • Immediate impact on practice [1256]<br>• Impact on policy & practice [478] |
| Quality | Scientific rigor and applicability of the articles published in the journal. | 7.8% | • Rigorous, robust, important, field changing, important, correct [488]<br>• High quality research [732] |
| Readership | Focuses on how much the published work is read by academic and non-academic people | 16.4% | • High readership, broad readership [154]<br>• Read widely [1171] |

Name, description, percentage of responses, and examples of the categories found in participants' definitions of High Impact. Numbers in square brackets represent anonymized identification of participants.

The analysis of the definitions provided for *prestige* resulted in 262 segments. These segments were coded under six categories: *Impact Factor and Metrics*, *Quality and Relevance*, *Readership*, *Relation to Associations*, *Reputation*, and *Review Process*. Table 2 provides definitions for each of the categories, as well as some examples for each of them.

The result of this coding process shows that the prestige of academic journals is, in a somewhat circular fashion, most commonly defined by their reputation, which is related to the name recognition of the journal, the people who publish in it or the people in charge of the review process. This was exemplified by definitions like "well-regarded by others in a field" [719], "the journal has a known name in my field of study" [1565] or "well regarded with global recognition" [827]. *Prestige* was also often defined based on Impact Factor and Metrics and by the Review Process used by the journal.

Finally, the analysis of the definitions provided for *high impact* resulted in 242 segments. These segments were coded under six categories: *Impact Factor and Metrics*, *Impact on Academia*, *Impact Outside Academia*, *Quality*, and *Readership*. Table 3 provides definitions for each of the categories, as well as some examples for each of them.

The result of this coding process shows that high impact of academic journals is defined by the JIF or other citation metrics in almost half of all instances. Definitions in this category included: "some factor that assess the impact" [581], "number of citations/ papers" [478] or "Interplay between Impact Factor and number of cites per year" [500]. To a lesser extent, *High Impact* was defined by the volume of readership the research receives and by the impact that the work had on practice, public policy, or in the media.

## Differences by demographics

We performed a Chi-Square test on the definitions of the three terms to understand if they varied depending on the gender, age, and academic discipline of the faculty member or according to the institution type to which they belong (R-type or M-type). The definitions provided by surveyed faculty do not have significant variation in any of these categories, which implies that

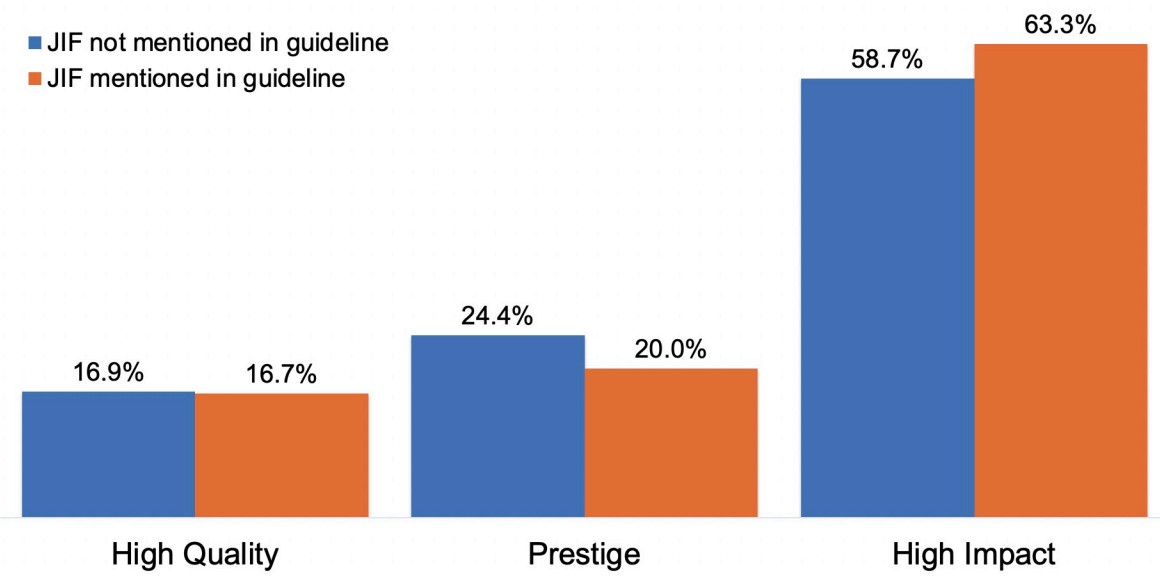

**Fig 1. Use of "Impact factor and metrics" as a definition of various terms.** Percentage of responses that contained at least one segment in participants' definitions of High Quality, Prestige and High Impact that relies on "Impact Factor and metrics", as a proportion of all the RPT guidelines that mentioned the JIF and of the guidelines that did not.

the academics conceive of these terms irrespective of their gender, age, academic discipline, or type of institution in which they are employed (See supplemental information).

## Differences by RPT guidelines

Finally, given the importance that academics give to the JIF and other metrics to define *high quality*, *prestige*, and *high impact*, we compared the responses received in the survey with the RPT documents from the academic units of the respondent. In particular, we performed a Chi-Square test to see if respondents used a definition coded as "Impact Factor and Metrics" for each of the terms any differently if they worked at academic units that mentioned the JIF and related terms in their RPT document. Fig 1 shows the prevalence of this definition among the two groups of faculty (those part of academic units that mention the JIF and those who do not). We do not find statistically significant differences between these groups [X2 = (5, N = 202) = 0.85, p > .05], indicating that the mention of the JIF and related terms in RPT documents does not affect how faculty define *high quality*, *prestige*, and *high impact*.

## Discussion

Our analysis of how faculty define quality, prestige, and impact of academic journals, suggests three important outcomes. First, it shows that these three terms, despite referring to three very different concepts, are often defined in overlapping ways (with references to each other and to themselves). Second, it shows that individual faculty members apply very different definitions to each of the terms, with no single definition used by over 50% of respondents for any of the three terms. Finally, the marked variance in definitions across faculty does not correspond to demographic characteristics, such as the age, gender, discipline nor to characteristics of the institution for which they work, including mentions of the JIF in their academic unit's RPT guidelines.

While it is known that there is a lack of definitions for many of the terms and concepts used in research assessment [18–21], this study explores how three key terms are understood by faculty in absence of these definitions. Specifically, our results indicate that the concepts of *quality*, *prestige*, *and impact* are often seen as synonymous by faculty, with *high quality* being sometimes defined by *reputation*, which is itself one of the most common definitions given for *prestige;* similarly, *high impact* is at times defined by *quality*, and *high quality* is at times defined by the impact factor. In fact, all three terms—*quality*, *prestige*, *and impact*—are defined through the use of the Impact Factor and other citation metrics by some faculty (with the term *Impact* itself defined in this way nearly half the time).

Given the subjective nature of these concepts and the overlapping definitions provided by faculty, it is perhaps unsurprising to see that, in all three cases, some faculty resort to the use of quantitative citation metrics as the basis for their definition. The rise in the use of metrics in research assessment has been well documented [44, 45], including in our own work that showed their prevalence in RPT documents and guidelines [10, 23]. In this sense, our study confirms that some faculty believe that *quality*, *prestige*, and *impact* of academic journals can be understood through citation metrics. However, contrary to our hypothesis that faculty would be more likely to think in terms of metrics if they were mentioned in the RPT guidelines for their institution, our study showed that respondents were no more or less likely to use citation metrics in their definitions when this was the case. Indeed, results of this study suggest that although RPT guidelines are meant to govern how quality, prestige, and impact are framed at universities, they might not have as much impact as intended among faculty. In other words, despite their importance in determining individual career advancement, the mention of the JIF in RPT guidelines is not correlated with how faculty define *quality*, *prestige*, or *impact*.

This, of course, only raises further questions about how faculty arrive at their understanding of *quality*, *prestige*, *and impact*. While our study does not offer direct answers to these questions, it does point to the wide range of definitions that are currently used by faculty when considering these career-determining aspects of research. Our study shows that in the absence of common definitions, faculty are applying their own understanding of each term and are doing so in ways that differ from their own colleagues, highlighting the subjectivity of the terms. This may stem from their own personal experiences with certain journals, reviewers, editors, and colleagues. For example, the highest percent of respondents in our survey perceived quality as being related to the review process of a journal. However, most journals do not make article reviews public, suggesting that the review process of a journal is not widely known or understood by people outside their own experiences. As it is widely documented that the peer review process can vary significantly, this highlights how personal experiences may affect how people perceive these different terms. This is precisely the situation that Hatch [20] warns could lead to biases (conscious or not) in evaluation processes and could explain in part why faculty are generally unsure of what is expected of them for tenure [28].

More broadly, our findings present a challenge for those seeking the most effective ways to bring about research evaluation reform. Unfortunately, our initial exploration suggests that the pathway for making changes to research assessment may not be as simple as clarifying how definitions are presented in assessment guidelines, given that the inclusion of metrics-related terms in RPT documents was not a determining factor in whether faculty used citation metrics in defining *high quality*, *high prestige*, or *impact*. This is not to say that such changes would not be worthwhile; guidelines, like policies and mandates, are an important way for departments and institutions to signal their values, even when those guidelines are not strictly adhered to. However, our research also suggests that determining how to define these specific terms would be challenging, given how subjectively faculty themselves define them. Overall, our research

points to the need for additional cultural and environmental factors that determine faculty thinking that cut across age, gender, institution types, and disciplines.

The reliance on metrics shown throughout this study further highlights two related issues. First, when it comes to understanding quality, prestige, and impact in evaluation, there is a tendency among faculty to gravitate towards definitions that facilitate comparisons between people and outputs, which given the subjectivity of such definitions, creates challenges for comparison especially across disparate disciplines. Second, in the search for such comparable measures, metrics like the JIF have come to be seen as a way of objectively assessing these qualities. These issues are problematic, in part because comparing measures of quality prestige and impact across research or individuals is itself questionable, as these concepts are context dependent, and in part because they fail to account for the numerous limitations and biases that are present in the creation and implementation of citation metrics [46–48].

In response to these issues, it is worth recognizing how efforts towards responsible use of metrics for research assessment—advanced by initiatives such as the San Francisco Declaration on Research Assessment (DORA) [34], Leiden Manifesto (n.d.), HuMetricSS (n.d.), the Hong Kong Principles [49] and narrative-based evaluations (see Saenen et al. [50] for a review)—may indeed help us to move away from the most problematic uses of metrics that are laden with the same challenges as ambiguous definitions. DORA advises against using any journal-level metrics, and the JIF in particular, as measures of research quality in evaluations and recommends instead relying on a variety of article-level metrics and qualitative indicators to better assess individual works; The Leiden Manifesto cautions against over-reliance on and incorrect uses of quantitative metrics, and emphasizes the importance of qualitative assessments as well as contextual and disciplinary considerations; HuMetricSS encourages the development of value-based assessments and proposes a definition of 'quality' as "a value that demonstrates one's originality, willingness to push boundaries, methodological soundness, and the advancement of knowledge."; The Hong Kong Principles recommend assessing research practices, such as complete reporting or the practices of open science, instead of citation-based metrics; and a number of funders and institutions worldwide are increasingly promoting the use of narrative approaches to allow researchers to more fully describe their work and its importance or influence in an efforts to decrease the reliance on metrics and avoid problematic quantitative comparisons altogether [50].

Complementing these initiatives, results of this study further highlight the need to implement evaluation regimes that don't rely on the comparisons of ill-defined concepts like those discussed here. Indeed, while we acknowledge the complexity of academic careers, our findings demonstrate that when terms such as "high quality", "prestige", and "high impact" are used in research and journal assessment, people perceive these differently across fields, contexts, and individual experiences. The use of such terms may lead academics in many directions—which may be quite desirable in terms of promoting a wide range of academic activities and outputs—but will likely lead to inconsistencies in how these activities are judged by RPT committees. Given the impossibility of universal definitions, it is understandable that many faculty fall back on measures that are comparable, but that, in reality, cannot capture the diversity of interpretations that exist within each context. As such, the findings of this study invite us to reconsider how, if at all, we want quality, prestige, and impact to be critical components of research assessment.

## Limitations

There are several limitations to the scope and interpretation of this work. First, the geographic focus area in Canada and the U.S. means that this work may not be representative of other

regions, especially of places without comparable academic rewards systems. As well, given the constraints of a brief survey instrument, we acknowledge that the answers reported may not fully reflect the nuance of how faculty understand the terms studied. Future research could better capture their understanding through other means, such as in-depth interviews. Finally, given that the survey utilizes self-reported information, we acknowledge that the definitions provided may not reflect the ones utilized by researchers when assessing academic journals. Future research could better connect individual responses with the results of evaluations that rely on the terms studied.

## Supporting information

**S1 Table. Breakdown of definitions by demographic characteristics.** *Overview of the participants' definition of Quality, Prestige and Impact by their gender, institution type, and age.* The color scale illustrates the distribution of responses, where green indicates a high percentage of responses and red indicates a low percentage of responses.
(DOCX)

**S2 Table. Breakdown of definitions by demographic characteristics.** *Overview of the participants' definition of Quality, Prestige, and Impact by discipline.* The color scale illustrates the distribution of responses, where green indicates a high percentage of responses and red indicates a low percentage of responses.
(DOCX)

## Author Contributions

**Conceptualization:** Erin C. McKiernan, Meredith T. Niles, Lesley Schimanski, Juan Pablo Alperin.

**Data curation:** Esteban Morales.

**Formal analysis:** Esteban Morales.

**Funding acquisition:** Erin C. McKiernan, Meredith T. Niles, Lesley Schimanski, Juan Pablo Alperin.

**Methodology:** Esteban Morales, Juan Pablo Alperin.

**Resources:** Juan Pablo Alperin.

**Supervision:** Juan Pablo Alperin.

**Visualization:** Esteban Morales, Juan Pablo Alperin.

**Writing – original draft:** Esteban Morales, Erin C. McKiernan, Meredith T. Niles, Lesley Schimanski, Juan Pablo Alperin.

**Writing – review & editing:** Esteban Morales, Erin C. McKiernan, Meredith T. Niles, Lesley Schimanski, Juan Pablo Alperin.

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
