## [Decision Letter · Decision Letter 0]

15 May 2021

PONE-D-21-10048

How faculty define quality, prestige, and impact in research

PLOS ONE

Dear Dr. Alperin,

Thank you for submitting your manuscript to PLOS ONE. After careful consideration, we feel that it has merit but does not fully meet PLOS ONE’s publication criteria as it currently stands. Therefore, we invite you to submit a revised version of the manuscript that addresses the points raised during the review process.

**First of all, I would like to thank the two reviewers.** They were fast and provided very insightful comments. One suggested to accept the manuscript and the second one suggested major revisions. In my opinion, the major revisions are doable and **I will be very pleased to assess this manuscript after your careful revisions** (based on the 2 reviewers' comments). Some additional comments : 

- In order to facilitate my assessment, please follow a reporting guideline (and please add a form of checklist). I suggest SRQR (https://www.equator-network.org/reporting-guidelines/srqr/) but I may be wrong so feel free to use any other guideline if you think that it fits better with your research. Thank you in advance. 

- Please also make it explicit in the method section if there was a pre-specified protocol for this very specific research question (the one presented in this paper) and it if it was registered (and where). Please attach the protocol in a supplementary file. If there was no protocol, nor registration, please make it explicit and justify. 

- Please add a few words in the text and abstract about the main limitations, to avoid any spin. 

We look forward to receiving your revised manuscript.

Kind regards,

Florian Naudet, M.D., M.P.H., Ph.D.

Academic Editor

PLOS ONE

Journal Requirements:

2. If materials, methods, and protocols are well established, authors may cite articles where those protocols are described in detail, but the submission should include sufficient information to be understood independent of these references (https://journals.plos.org/plosone/s/submission-guidelines#loc-materials-and-methods).

Reviewers' comments:

Reviewer's Responses to Questions

**Comments to the Author**

1. Is the manuscript technically sound, and do the data support the conclusions?

Reviewer #1: Yes

Reviewer #2: Yes

2. Has the statistical analysis been performed appropriately and rigorously? 

Reviewer #1: N/A

Reviewer #2: Yes

3. Have the authors made all data underlying the findings in their manuscript fully available?

Reviewer #1: Yes

Reviewer #2: Yes

4. Is the manuscript presented in an intelligible fashion and written in standard English?

Reviewer #1: Yes

Reviewer #2: Yes

5. Review Comments to the Author

Reviewer #1: Review of PONE-D-21-10048

This manuscript reports on the answers to an open survey question about how faculty of universities in the USA and Canada would define quality, prestige and impact of academic journals. The article is interesting and well-written but a number of major and minor concerns should be adequately responded to with a view to optimize clarity and relevance of the manuscript.

Major concerns

It’s not very clear what quality, prestige and impact refer to: research (as in the title), journals (as asked in the survey), to the publication oeuvre of an individual researcher (as one would expect given the focus on review, promotion, and tenure (RPT) processes), or to all of these categories combined (which would be strange and confusing). Please clarify and apply the choice consistently throughout the manuscript.

I agree that having a clear definition or description of core concepts used in the assessment of research and researchers is a necessary starting point but please also discuss the following.

o The meta-question whether quality, prestige and impact are the main concepts we need to look at. You allude to this a bit in the Introduction section but remain silent in the Discussion section and the Abstract.

o The practical question how quality, prestige and impact can be operationalized. This seems to be the main focus of your respondents (and less so the conceptual definitions).

Maybe also mention (in the Discussion section) the best practices presented by DORA on their website and the recently introduced Hong Kong principles:

Moher D, Bouter L, Kleinert S, Glasziou P, Sham MH, Barbour V, Coriat AM, Foeger N, Dirnagl U. The Hong Kong principles for assessing researchers: fostering research integrity. PLoS Biology 2020; 18: e3000737. (https://journals.plos.org/plosbiology/article?id=10.1371/journal.pbio.3000737) (translated in Chinese, German, Portuguese)

I’m frankly puzzled that both the documents you analysed earlier and the faculty you surveyed don’t focus much more on the H-index. Although this measure – which is frequently used in researcher assessments in Europe – is also deeply flawed it’s clearly superior to the JIF for assessing researchers for two reasons: 1) the H-index concerns the whole publication oeuvre of an individual, and 2) the H-index is based on the actual citations to the articles the researcher at issue published. Please comment on this in the Discussion section.

It’s not so clear to me why you would expect differences between subgroups defined by demographic characteristic and why that would be interesting or relevant to know.

It’s also not clear why you would expect a high correlation between the views of faculty and the policy documents in their university. On a slightly cynical note: who is reading these documents? My guess is hardly anyone.

The manuscript is much longer than necessary and should be shortened substantially by e.g.

o Removing lines 16-30 on page 5 and lines 1-21 on page 6. This lengthy description of earlier work doesn’t belong in the Methods section (or elsewhere in the article) and can be replaced by one or two sentences that only explain what’s really of direct importance to understand the methods of what is reported in this manuscript.

o Figures 1-3 have little informative value and can easily be removed when the percentages reported are transferred to tables 1-3.

o Table 4 and 5 can better be moved to the digital supplements as their message (‘no significant differences between subgroups’) is already well described in two lines in the text.

Minor concerns

What was the response of the survey? It’s unclear how many invitations were sent. And how many participants answered the open question on which this manuscript is based? Respondents often skip open questions. Please produce a flow chart.

Was your survey pre-registered? Please clarify and add the pertaining link if you did.

Please keep the order of quality, prestige and impact uniform throughout the text and also order tables 1, 2 and 3 likewise.

In table 1 you say ‘definitions of High Quality’ in the title but talk about ‘definitions of High Impact in the note directly below the table. That last formulation seems to be wrong.

It’s not clear what the different colours in table 4 and 5 indicate.

I sign my reviews:

Lex Bouter, Amsterdam University Medical Centers and Vrije Universiteit Amsterdam

Reviewer #2: The research shows that absolute general definitions of quality, impact and prestige which may be used to compare researchers and research within disciplines and cross academia may be difficult if not impossible to define. It is also because of this that in academia we started to use 'objective' indicators and metrics which have been shown however to be poor proxies for quality and excellence. In the attempt to go beyond these flawed metrics there are now many initiatives to design meaningful ways to evaluate research. Based on their observation in this paper, the authors feel that is problematic since quality, excellence and prestige are ill-defined and researchers have many different views of it and most times rely on the flawed classical metrics, i.e. papers and where they are published and cited.

I would suggest to the authors to include a discussion of those initiatives that really are trying to deal with this problem. In the introduction they cite several of the papers that discuss this, f.i . Aksnes et al 2019, Hicks et al 2015 which is The leiden manifesto and Moher et al 2018.

Instead of looking for the impossible, i.e. absolute, timeless measures for quality and prestige, these initiatives start with the conclusion that quality, excellence and prestige are context dependent. Research quality, its products and evaluation are highly context dependent which makes direct comparisons of historians, philosophers and chemists and even researchers within for instance the domain of biomedical and health research, irrelevant. It is about 'rigor, plausibility, originality, societal value' (Asknes 2019) but in a given research setting, with a strategy, aim, specific goals, a process (f. i .Open Science practices) and if applicable actions in the corresponding societal context (Nowotny et al, Rethinking Science, 2001).

To say that research is Excellent or Good requires from peers/reviewers, a narrative, a motivation of judgement of strengths and weakenesses based on reading and understanding of the content of the work.

This is the approach taken in the recently adopted National Strategic Evaluation Protocol (SEP) in The Netherlands

https: //www.vsnu.nl/files/documenten/Domeinen/Onderzoek/SEP_2021-2027.pdf

6. PLOS authors have the option to publish the peer review history of their article (what does this mean?). If published, this will include your full peer review and any attached files.

Reviewer #1: **Yes: **Lex Bouter, Amsterdam University Medical Centers and Vrije Universiteit Amsterdam

Reviewer #2: **Yes: **Frank Miedema

---

## [Author Response · Author response to Decision Letter 0]

2 Jul 2021

Dear Editor and Reviewers, 

Thank you and the two reviewers for the helpful feedback on our manuscript “How faculty define quality, prestige, and impact in research”. We are pleased to resubmit our manuscript with the requested revisions. Attached you will find the reviewers’ feedback (in gray) interspersed with a description of how we addressed each of the points raised (in black). 

We would like to take this opportunity to express our appreciation to the reviewers for their thoughtful feedback. We are convinced that we have adequately addressed all the expressed concerns and that the manuscript has been improved as a result of this process.

Sincerely, 

Juan Pablo Alperin

Assistant Professor, Publishing

Associate Director, Public Knowledge Project

Director, Scholarly Communications Lab

Simon Fraser University

---

## [Editor Report · Decision Letter 1]

7 Jul 2021

PONE-D-21-10048R1

How faculty define quality, prestige, and impact of academic journals

PLOS ONE

Dear Dr. Alperin,

Thank you for submitting your manuscript to PLOS ONE. After careful consideration, we feel that it has merit but does not fully meet PLOS ONE’s publication criteria as it currently stands. Therefore, we invite you to submit a revised version of the manuscript that addresses the points raised during the review process.

Thank you for answering the reviewers's comments. However, after a rapid check, I'm afraid that you may have missed some of my editorial points. Please excuse me if I am wrong. 

- In order to facilitate my assessment, please follow a reporting guideline (and please add a form of checklist). I suggest SRQR (https://www.equator-network.org/reporting-guidelines/srqr/) but I may be wrong so feel free to use any other guideline if you think that it fits better with your research. Thank you in advance. 

- Please also make it explicit in the method section if there was a pre-specified protocol for this very specific research question (the one presented in this paper) and it if it was registered (and where). Please attach the protocol in a supplementary file. If there was no protocol, nor registration, please make it explicit and justify. 

- Please add a few words in the text and abstract about the main limitations, to avoid any spin. 

We look forward to receiving your revised manuscript.

Kind regards,

Florian Naudet, M.D., M.P.H., Ph.D.

Academic Editor

PLOS ONE
---

## [Decision Letter · Decision Letter 2]

26 Jul 2021

PONE-D-21-10048R2

How faculty define quality, prestige, and impact of academic journals

PLOS ONE

Dear Dr. Alperin,

Thank you for submitting your manuscript to PLOS ONE. After careful consideration, we feel that it has merit but does not fully meet PLOS ONE’s publication criteria as it currently stands. Therefore, we invite you to submit a revised version of the manuscript that addresses the points raised during the review process.

**First of all, I would like to thank the 2 reviewers for their fast peer review. As you will see there are still 2 minor issues. **I agree with one reviewer that the 2 tables can be moved in appendix. I suggest to follow this suggestion. If you disagree, please explain why these tables/figures are, in your view, necessary. There is also a last conceptual point from the other reviewer and I think that he makes a good point here. Please address his comment.

Pending these two minor changes/edits, I'll be more than happy to accept this manuscript for publication. 

We look forward to receiving your revised manuscript.

Kind regards,

Florian Naudet, M.D., M.P.H., Ph.D.

Academic Editor

PLOS ONE

Journal Requirements:

Reviewers' comments:

Reviewer's Responses to Questions

**Comments to the Author**

1. If the authors have adequately addressed your comments raised in a previous round of review and you feel that this manuscript is now acceptable for publication, you may indicate that here to bypass the “Comments to the Author” section, enter your conflict of interest statement in the “Confidential to Editor” section, and submit your "Accept" recommendation.

Reviewer #1: All comments have been addressed

Reviewer #2: (No Response)

2. Is the manuscript technically sound, and do the data support the conclusions?

Reviewer #1: Yes

Reviewer #2: Yes

3. Has the statistical analysis been performed appropriately and rigorously? 

Reviewer #1: Yes

Reviewer #2: I Don't Know

4. Have the authors made all data underlying the findings in their manuscript fully available?

Reviewer #1: Yes

Reviewer #2: Yes

5. Is the manuscript presented in an intelligible fashion and written in standard English?

Reviewer #1: Yes

Reviewer #2: Yes

6. Review Comments to the Author

Reviewer #1: (No Response)

Reviewer #2: My comment:

"Instead of looking for the impossible, i.e. absolute, timeless measures for quality and prestige,

these initiatives start with the conclusion that quality, excellence and prestige are context

dependent. Research quality, its products and evaluation are highly context dependent which

makes direct comparisons of historians, philosophers and chemists and even researchers within

for instance the domain of biomedical and health research, irrelevant. It is about 'rigor,

plausibility, originality, societal value' (Asknes 2019) but in a given research setting, with a

strategy, aim, specific goals, a process (f. i .Open Science practices) and if applicable actions in

the corresponding societal context (Nowotny et al, Rethinking Science, 2001).

To say that research is Excellent or Good requires from peers/reviewers, a narrative, a motivation

of judgement of strengths and weakenesses based on reading and understanding of the content of

the work.

This is the approach taken in the recently adopted National Strategic Evaluation Protocol (SEP)

in The Netherlands"

https: //www.vsnu.nl/files/documenten/Domeinen/Onderzoek/SEP_2021-2027.pdf

The author's response:

We very much agree with this reviewer’s view and believe that the research presented here

(alongside the other publications from this project) help to confirm this perspective by showing,

through concrete empirical evidence, some of the pernicious effects of using seemingly objective

measures and standard definitions for evaluations that should be context specific

My response now:

Despite this positive reaction , the author's in the paper do not respond to this major comment of mine. They write in the paper: "While it is known that there is a lack of definitions for many of the terms and concepts used in research assessment (Dean et al., 2016; Hatch, 2019; Moore et al., 2017; van Mil & Henman, 2016), this study explores how three key terms are understood by faculty in absence of these definitions."

They leave it open at the end of the discussion how to deal with this problem. It is felt to be a problem by those who believe it useful to compare evaluations between very different fields of research. These comparisons now based on JIF, h-index. numbers of citations etc are as they argue correctly deeply flawed. They do not conclude, as they seem to do in the reply to my comments, that these terms are dependent on discipline, sub-disciplines and strategy and on thematics of research fields and topics. Give the change in science and society they are not timeless either.

Thus, that many terms they have investigated are quite different in use and have no absolute universal definition or meaning, must mean that looking for absolute generally applicable indicators is not the way to approach what by many is felt to be a problem. This inescapable conclusion, which for many brought up in current science may be difficult, should be mentioned in the discussion and that, as a way forward, we need to broadly introduce and train the use of narratives both from researchers and reviewers, auditors, peers, to transparently deal with this context dependency. This is possible impact of their work on development of policies of research evaluations and highly relevant to suggest.

The SEP in The Netherlands has dealt with this 'problem', but narratives have been introduced in the REF in the UK and elsewhere before.

7. PLOS authors have the option to publish the peer review history of their article (what does this mean?). If published, this will include your full peer review and any attached files.

Reviewer #1: **Yes: **Lex Bouter, professor of Methodology and Integrity, Amsterdam University Medical centers and Vrije Universiteit Amsterdam, The Netherlands

Reviewer #2: **Yes: **Frank Miedema

---

## [Author Response · Author response to Decision Letter 2]

26 Aug 2021

Thank you and the two reviewers for this rapid second review. We apologize for our own delay in returning our manuscript “How faculty define quality, prestige, and impact in research”. The response came while many of us were away on vacation and it is only now that we were all able to review our response. 

We are pleased to resubmit our manuscript with the requested revisions. As you can see in the attached manuscript, we have moved the two tables to supplementary materials and made substantial additions to directly address the remaining reviewer comment in the discussion. You will now find additions that directly include the reviewer’s suggestions on p. 13 (lines 15-16; 19-29), p. 14 (lines 13-17; 22-28), as well as in a line in the abstract. 

We are certain these additions would be met with agreement by the reviewer, as they substantially incorporate their views into the discussion and conclusions of our work. We trust you will agree.

---

## [Decision Letter · Decision Letter 3]

31 Aug 2021

How faculty define quality, prestige, and impact of academic journals

PONE-D-21-10048R3

Dear Dr. Alperin,

We’re pleased to inform you that your manuscript has been judged scientifically suitable for publication and will be formally accepted for publication once it meets all outstanding technical requirements.

**I would like to thank you for all the work you did during the peer review process and I would like to thank the 2 reviewers again for their important feedback.**

Kind regards,

Florian Naudet, M.D., M.P.H., Ph.D.

Academic Editor

PLOS ONE

Additional Editor Comments (optional):

Reviewers' comments:

Reviewer's Responses to Questions

**Comments to the Author**

1. If the authors have adequately addressed your comments raised in a previous round of review and you feel that this manuscript is now acceptable for publication, you may indicate that here to bypass the “Comments to the Author” section, enter your conflict of interest statement in the “Confidential to Editor” section, and submit your "Accept" recommendation.

Reviewer #2: All comments have been addressed

2. Is the manuscript technically sound, and do the data support the conclusions?

Reviewer #2: Yes

3. Has the statistical analysis been performed appropriately and rigorously? 

Reviewer #2: N/A

4. Have the authors made all data underlying the findings in their manuscript fully available?

Reviewer #2: Yes

5. Is the manuscript presented in an intelligible fashion and written in standard English?

Reviewer #2: Yes

6. Review Comments to the Author

Reviewer #2: (No Response)

7. PLOS authors have the option to publish the peer review history of their article (what does this mean?). If published, this will include your full peer review and any attached files.

Reviewer #2: **Yes: **Frank Miedema

---

## [Editor Report · Acceptance letter]

18 Oct 2021

PONE-D-21-10048R3 

How faculty define quality, prestige, and impact of academic journals 

Dear Dr. Alperin:

I'm pleased to inform you that your manuscript has been deemed suitable for publication in PLOS ONE. Congratulations! Your manuscript is now with our production department. 

Kind regards, 

on behalf of

Pr. Florian Naudet 

Academic Editor

PLOS ONE